# OpenReview forum: "UnGuide: Learning to Forget with LoRA-Guided Diffusion Models"
_ICLR.cc/2026/Conference — Submitted to ICLR 2026_

### Official Review · Reviewer_Y2JL · 2025-10-29

**Soundness:** 2
**Presentation:** 2
**Contribution:** 2
**Rating:** 2
**Confidence:** 4

**Summary:**

The paper proposes a new unlearning method using LoRA. Prior LoRA-based approaches tended to push the target concept outside the data manifold or degrade the model’s ability to generate non-erasing concepts. To address this, the authors clarify the mapping concept so that the content replacing the erased concept is explicitly defined, and leverage an AutoGuidance-inspired strategy to further enhance unlearning relative to the base model. Additionally, the method automatically determines whether the erasing target appears in the prompt and dynamically adjusts the guidance scale, improving both usability and performance.

**Strengths:**

- **Simple and intuitive writing:** The authors clearly identify limitations of existing methods and effectively address them via an automated dynamic guidance-scaling strategy. Applying guidance between the base model and the LoRA model and adjusting the scale dynamically is intuitive and yields strong results.
- **Strong performance:** Under the benchmarks selected by the authors, the proposed method demonstrates distinctly superior performance over prior methods.

**Weaknesses:**

- **Limited novelty:** The method largely appears to extend AutoLoRA[1] to an unlearning LoRA model and a base model, with dynamic guidance scaling being the primary novel contribution. This raises concerns about whether the level of novelty is sufficient.
- **Dependence on mapping content:** As the authors note, avoiding the OoD manifold seems achievable because the mapping content offers a clear alternative for generation after erasing. However, this introduces strong dependence on mapping content, reduces diversity by collapsing the outcome into a single alternative, and imposes overhead on selecting a good mapping concept. While this might be manageable for settings like CIFAR-10 with few candidate classes, it is unclear how mapping content should be selected in a large-scale scenario. The paper provides no convincing rationale or criteria for this selection.
- **Limited experimental scope:** SD-v1.4 is now considered outdated. It is unclear whether the method generalizes to SD-XL or more recent TTI models such as SD3 or FLUX. Evidence of applicability to these models would demonstrate greater generality. Additionally, CIFAR-10 has only 10 classes, making it a limited benchmark for concept erasing. Although the paper appears to adopt benchmarks used in MACE [2], MACE also includes erasing across ~200 celebrity identities; including such experiments would strengthen claims about scalability to numerous and diverse concepts.
- **Motivation for AutoGuidance-based formulation:** The authors emphasize that borrowing the AutoGuidance structure improves stability. While it is understandable that it may help strengthen the unlearning effect relative to the base model, the rationale for why it specifically enhances stability is not clearly explained. Stronger motivation or justification is needed.
- **Disorganized presentation:** While the writing itself is reasonably easy to follow, the figures are poorly placed and disrupt the flow. Reorganizing figure placement to follow the narrative more closely would greatly improve readability.

[1] AutoLoRA: Automatically Tuning Matrix Ranks in Low-Rank Adaptation Based on Meta Learning, Zhang et al., 2024

[2] MACE: Mass Concept Erasure in Diffusion Models, Lu et al., 2024

**Questions:**

In line 134, “,” should be replaced with “.” as the end of a sentence.

---

> ### Author Response · Authors · 2025-11-27
>
> **W1**: AutoLoRA [1] addresses a very different problem: it focuses on automatically choosing LoRA ranks using meta-learning, whereas our work tackles concept unlearning in diffusion models and introduces an adaptive guidance mechanism at inference time. AutoLoRA does not perform unlearning, does not compare base and adapted models, and does not use any form of guidance modulation. Thus, although both methods involve LoRA, the goals and techniques are fundamentally different. We believe the reviewer was referring to the AutoLoRA [3] method. Our approach is inspired by their idea, but extends it with an additional statistical analysis of norm values, which enables selecting the guidance strength in a more principled and stable manner.
>
> [1] AutoLoRA: Automatically Tuning Matrix Ranks in Low-Rank Adaptation Based on Meta Learning, Zhang et al., 2024
>
> [3] AutoLoRA: AutoGuidance Meets Low-Rank Adaptation for Diffusion Models, Kasymov et al., 2024
>
> **W2**: Thank you very much for this helpful suggestion. Our goal was to follow a setting similar to MACE, where a mapping concept is also used during unlearning. We therefore adopt the same design choice to ensure comparability with prior work, but we agree that exploring automated or large-scale mapping selection is an interesting direction for future research.
>
> **W3**: We selected SD-1.4 and CIFAR-10 to ensure controlled comparison with prior unlearning works such as MACE. Extending the method to SD-XL and larger concept sets is an important and natural next step, and we plan to include such experiments in the future.
>
> Unlearning 100 concepts is a complex task, so in MACE, the researchers used the remaining knowledge for unlearning, as the model tends to over-forget. Therefore, we conducted additional experiments for unlearning 100 painting styles, where the UnGuidance mechanism was added to the MACE model. We obtained the following results: $\text{CLIP}_e = 20.64$, $\text{CLIP}_s = 28.65$, which gives $H_a = 8.01$ (where MACE obtained $H_a = 5.99$), improving the model's performance in distinguishing unlearned and remaining styles. Additional visualizations are presented in Fig. 9.
>
> **W4**: Thank you for pointing out the need for a clearer motivation of the AutoGuidance-based formulation. UnGuide adopts an AutoGuidance-style combination of the base and LoRA-adapted models because this structure explicitly separates two complementary roles: (i) the base model anchors the denoising trajectory close to the original data manifold, and (ii) the LoRA-adapted model injects a “repulsive” signal that enforces forgetting of the target concept. By interpolating these two conditional predictions in the noise space, UnGuidance constrains the effect of LoRA to a controlled direction rather than allowing the LoRA-adapted model to drive the entire trajectory, which empirically reduces unstable, off-manifold generations. Prior work on AutoGuidance and AutoLoRA shows that such dual-branch guidance improves both robustness and sample quality by correcting biased or undertrained variants of the model with a stronger reference model. In UnGuide, the same mechanism is repurposed for unlearning: the base model provides a stable reference, while the LoRA branch enforces concept removal, and their weighted combination yields more stable trajectories than using the LoRA-adapted model alone, as reflected in our higher $\text{H}_o$ and CLIP/FID scores relative to other unlearning baselines.
>
> Building on the discussion above, we have incorporated appropriate clarifications and enhancements in Section 3 of the revised paper.
>
> **W5**: In the revised version, we improved the organization of the manuscript by repositioning several figures so that they appear closer to the corresponding parts of the text.
>
> **Q1**: In the revised version, we have made an appropriate correction.

---

### Official Review · Reviewer_uC8E · 2025-10-29

**Soundness:** 2
**Presentation:** 3
**Contribution:** 2
**Rating:** 2
**Confidence:** 3

**Summary:**

The author's propose UnGuide, a method for improving LoRA based unlearning methods in diffusion models. In particular, the authors address how base LoRA methods inadvertently alter content unrelated to the desired erased concepts. To do this, they adaptively switch between the base and LoRA models during inference: boosting the LoRA model on prompts related the erased concepts, and boosting the base model on prompts unrelated to the erased concepts. They provide experiments showing their method achieves better targeted removal than other LoRA-based methods.

**Strengths:**

- The experimental results are strong, showing clear advantages over prior methods
- The method is also much simpler than the strongest prior methods, as it does not require external segmentation components.

**Weaknesses:**

- The paper lacks ablations comparing their method to a base LoRA approach i.e. without using un-guidance. The authors state the key component of the work is the dynamic switching between the base and LoRA models, but at least as I can see this is not ablated in the experiments. It would greatly strengthen the insights obtained from the paper if this could be included (or make it more prominent in case I missed it accidentally)
- I also could not find an explicit formula for w. Is it just a binary switch between 1 and -1 depending on which side of the threshold it is?
- Table 8 provides average values for the norms of the deltas, but distribution-level results should also be provided i.e. what is the error rate of the test?
- Lastly, that I can see the paper also lacks ablations for adversarially chosen prompts.

While my score recommends reject, I would be happy to raise it if these ablations and my questions below are addressed.

**Questions:**

- Would it be possible to include ablations as mentioned in the weaknesses question?
- I am also confused as to how UnGuide avoids distorting unrelated concepts. If I understood it right, the switching between the base and LoRA models is controlled by the degree to which they disagree on the outputs. Thus, if the LoRA model wrongly alters an unrelated concepts, causing the two models to disagree, UnGuide would still boost the LoRA model.
- What is the exact formula for w? Does it depend on the time-step or is it fixed for all generation steps?

---

> ### Author Response · Authors · 2025-11-27
>
> **W1**: To demonstrate the impact of UnGuidance, we present additional results for unlearning automobile and cat concepts. Our mechanism allows for better $\text{H}_o$ results than just the unlearned model (only LoRA adapter), for example, for a car it is a change from 76.98\% to 96.91\%, and for a cat from 48.82\% to 97.71\% (see Section 4).
>
>
> **W2/Q3**: To clarify how the guidance weight $w$ is assigned, we added an explicit sentence to the Dynamic Adaptation of Guidance Scale Section. The added line states that we set  $w\geq 1$ when the mean norm falls below the mean norm of the empty prompt and $w\leq −1$ when it exceeds it. This makes the decision rule for $w$ explicit.
>
>
> **W3**: Thank you for highlighting the potential error in mean norm calculations. In the revised version of the paper, we have added a new section (Analysis of Decision Reliability) within the ablation study, where we analyze the behavior of norm statistics. For each class, we computed the accuracy, defined as the proportion of repetitions (out of 50) in which the mean norm for that class appeared on the correct side of the neutral prompt value. The results of this experiment are presented in Tab. 3.
>
> **W4**: We evaluated the model on adversarial prompts related to the removed “cat” concept. The model handles these challenging prompts, as shown in Tab.10 (list of prompts) and Fig.7 (visual results).
>
> **Q1**: We thank the Reviewer for these helpful suggestions. We have made the revisions and addressed the weaknesses.
>
> **Q2**: We acknowledge that such a phenomenon may occur. This is precisely why our method requires a cleaner and more targeted unlearning stage using LoRA fine-tuning before applying UnGuidance. In addition, during UnGuidance, we compute a mean norm-based measure that is robust to such outlier concept shifts. We consider that these two steps together effectively minimize the risk of mistakenly reinforcing unwanted alterations in unrelated concepts.

---

### Official Review · Reviewer_K6aQ · 2025-10-31

**Soundness:** 2
**Presentation:** 2
**Contribution:** 2
**Rating:** 2
**Confidence:** 3

**Summary:**

The paper proposes UnGuide, a LoRA-guided diffusion unlearning method that adaptively balances outputs from a pretrained base model and a LoRA adapter. Unlike prior LoRA fine-tuning that unintentionally harms unrelated concepts, UnGuide dynamically adjusts a guidance scale w based on the early denoising variance. High-variance trajectories (indicating the presence of the forgotten concept) receive negative guidance to steer sampling back to the data manifold, while low-variance ones use positive guidance to preserve fidelity.
Experiments on object erasure and explicit content removal show that UnGuide selectively removes targeted concepts with minimal degradation in visual quality, outperforming existing LoRA-based unlearning baselines.

**Strengths:**

The work is original in its formulation of adaptive guidance for unlearning. While prior methods such as MACE rely on complex prompt or segmentation modifications, UnGuide innovatively combines a standard LoRA fine-tuning setup with a new variance-based guidance mechanism to dynamically balance between forgetting and fidelity.

**Weaknesses:**

The paper presents an interesting direction but suffers from several issues in organization, clarity, and experimental rigor that obscure its true contributions and weaken its overall impact.

* **Poor organization and unclear flow:** The paper’s structure makes it difficult to distinguish between background material and novel contributions. For instance, the description of the *Text-to-Image generation framework*—a preliminary concept—is embedded directly in the *Methodology* section, and is immediately followed by *LoRA for Unlearning* without clear separation. As a result, it is hard for readers to identify where the authors’ original ideas begin. Even within the LoRA section, preliminaries and proposed components are intermixed, and the objective of LoRA training is never clearly introduced before detailing its implementation specifics (e.g., use of predefined target prompts, reliance on the model’s intrinsic capabilities, and updates restricted to Key/Value matrices). The section starting from L240 could be significantly improved by clearly stating the **training objective first**, followed by implementation details.

* **Ambiguity in novelty claim:** The LoRA training procedure itself is conceptually equivalent to *distilling negative prompt guidance* into a student model, an idea already explored in **ESD [1]** and related works. Those methods fine-tuned the full model rather than using LoRA adapters, but this distinction is largely an implementation convenience rather than a novel algorithmic contribution. However, the paper frames this distillation-based objective as an original idea, which overstates its novelty.

* **Unclear and computationally expensive adaptive guidance:** The paper’s main claimed contribution—the *adaptive inference-time guidance*—relies on repeatedly computing the difference between noise predictions of the base and unlearned models to determine whether a prompt contains a forgotten concept. This process requires averaging over 10–30 stochastic samples, which can increase inference cost by an order of magnitude. The paper does not convincingly demonstrate that this heavy computation yields proportional gains in quality or controllability. Furthermore, when (w > 1) (i.e., the prompt is judged not to contain the forgotten concept), it is unclear why extrapolating **away from** the unlearned model’s predictions is necessary—logically, the base model alone should suffice. The motivation for this design choice is missing.

* **Inconsistency in guidance formulation:** The authors state that in practice (w = -1) or (w = 2) (L649), even though Eq. (5) defines (w = 0.5) as the balanced midpoint between models. Therefore, the earlier notation (w < -1) or (w > 1) (L282, L291) is inconsistent with the center defined at 0.5. It would be more natural to express the condition as (w < -1) or (w > 2) to align with the intended midpoint. This inconsistency makes it hard to interpret how (w) influences the final prediction.

* **Lack of analysis for key scaling parameters (γ and w):** In diffusion guidance literature, the guidance scale is known to have a major impact on sample quality—too high causes oversaturation, too low leads to weak effects. Most prior works analyze performance sensitivity to this scale. However, the paper provides no ablation or sensitivity analysis for its two main scaling parameters, (γ) (training repulsion strength) and (w) (inference guidance weight). Without such analysis, it is difficult to assess the stability or robustness of the method.

Overall, the work introduces an interesting adaptive idea but would benefit greatly from a clearer exposition of its contributions, justification for design choices, and more systematic experiments—especially around guidance scale behavior and computational efficiency.

[1] Erasing Concepts from Diffusion Models, Rohit Gandikota, Joanna Materzynska, Jaden Fiotto-Kaufman, David Bau

**Questions:**

**Questions:**

- Is there a clear justification for why the model needs to interpolate between the unlearned model and the base model at inference time? Why not simply use one of them depending on whether the prompt contains the forgotten concept?
- Why is extrapolation beyond the base model ((w > 1)) necessary when the prompt does not contain the forgotten concept? Would simply using the base model in such cases yield similar or better results?
- What is the computational overhead of averaging over 10–30 stochastic samples to estimate LoRA influence? Could a more efficient proxy be used without sacrificing adaptivity?
- How sensitive is the method to the choice of the key scaling parameters (γ) (training repulsion strength) and (w) (guidance weight)? An ablation or stability analysis could strengthen the claims.
- Can the authors better distinguish their LoRA training objective from prior negative-guidance distillation methods such as ESD [1]? What conceptual or empirical improvement does UnGuide introduce beyond replacing full fine-tuning with LoRA adapters?
- Can the authors provide an ablation study comparing results when: (a) only the base model is used, (b) only the unlearned model is used, and (c) their proposed adaptive guidance is applied, to show the quality and unlearning trade-off?

---

* **My general take:** The idea of removing unwanted influence from prompts unrelated to the unlearned concept is interesting, but the current method feels underdeveloped. The central question is whether the cost of repeatedly estimating model differences can be justified — either by improving its efficiency or by demonstrating that the benefit is significant enough to warrant this cost. Showing an ablation that quantifies the efficiency and necessity of guidance between the unlearned and base models (the paper’s core contribution) would be essential. I would welcome clarification if I have misunderstood this aspect, and a convincing explanation or experiment could positively affect my rating.

---

> ### Author Response · Authors · 2025-11-27
>
> **W1**: We thank the reviewer for the helpful comment. Following the suggestion, we added a clear sentence at the start of the LoRA for Unlearning subsection to state the training objective. This makes it easier to see where our contribution begins and improves the clarity of the Methodology section.
>
> **W2/Q5**: Our goal was not to introduce a new LoRA training procedure, but rather to utilize existing ideas, such as ESD, to create the unlearned model required for UnGuidance. The mapping concept we employed extends the ESD approach and offers a more stable signal than using an empty prompt. The main innovation of our work is the UnGuidance mechanism, which adaptively combines the base model with the unlearned model, ensuring stable and controlled unlearning.
>
> **W3/Q2**:  As in classifier-free guidance, where users may increase the guidance scale (usually 7.5, and even 20) even though the default value is typically the most stable, our parameter $w \geq 1$ acts as a controllable hyperparameter. In practice, $w = 1$ is the natural setting, but allowing slightly larger values (e.g., $w = 2$) provides additional flexibility while still keeping the sampling trajectory close to the base model for non-target prompts.
>
> **W4**: We want to clarify that while Equation (5) suggests a midpoint at $w = 0.5$, in our method, w is not meant to be used as a symmetric interpolation parameter. Instead, we set the condition $w >= 1$ to ensure that sampling follows the base model. Any value that meets $w >=1$ is valid, and we used  $w = 2$ as an example. This choice does not suggest a different midpoint; it simply offers a way for users to control the guidance strength, which is consistent with standard diffusion practices.
>
> **W5/Q4**: We calculated results for different values of $\gamma$, showing that the training remains different across a range of repulsion strengths. We also clarified the LoRA for Unlearning section by more clearly separating background from our contribution.
>
> We performed additional ablation focusing on two classes (cat and automobile) and calculated metrics for $\text{Acc}_e$, $\text{Acc}_s$, $\text{Acc}_g$, and $\text{H}_o$. The results (see Tab. 4) show that the method is stable across the entire range; for $\gamma=1$ or $\gamma=2$ the effect is similar, but for $\gamma=3$ the model may forget too much, which results in a low $\text{Acc}_s$ metric, as in the situation for automobile.
>
> Table for cat:
> | $\gamma$ | $\text{Acc}_e$ ↓ | $\text{Acc}_s$ ↑ | $\text{Acc}_g$ ↓ | $\text{H}_o$ ↑ |
> |------------|---------------------|--------------------|--------------------|-----------|
> | 1          | 2.43                | 98.62              | 4.34               | 97.27     |
> | 2          | 2.98                | 98.80              | 2.66               | 97.71     |
> | 3          | 2.25                | 98.55              | 3.55               | 97.58     |
>
> Table for automobile:
> | $\gamma$ | $\text{Acc}_e$ ↓ | $\text{Acc}_s$ ↑ | $\text{Acc}_g$ ↓ | $\text{H}_o$ ↑ |
> |------------|---------------------|--------------------|--------------------|-----------|
> | 1          | 1.45                | 98.05              | 5.82               | 96.89     |
> | 2          | 1.83                | 97.95              | 5.32               | 96.91     |
> | 3          | 1.40                | 88.04              | 2.30               | 94.53     |
>
> **W3/W6/Q3**: The computational overhead is low: we use a single denoising trajectory up to step $t$ (typically $25\leq t \leq 50$) on the base model and then evaluate one noise prediction per model. As shown in Tab. 11, the norm-gap statistic stabilizes already for $\text{repeats} \geq 10$. Moreover, partially denoised latents (Fig. 15-16) are reused as the starting point for final generation, so no second denoising pass is required.
>
> **Q1**: When the prompt contains the forgotten concept, simply using the unlearned model is often too weak, so additional repulsion is needed to push the sampling trajectory further away from that concept. This is why we apply $w \leq −1$. Conversely, when the prompt is unrelated, $w \geq 1$ keeps the generation close to the base model. This interpolation mechanism thereby strengthens forgetting where needed while preserving overall image quality elsewhere.
>
> **Q6**: We want to clearly demonstrate the impact of using UnGuidance.  We present additional results for unlearning automobile and cat concepts. Our mechanism allows for better $\text{H}_o$ results than just the unlearned model (only LoRA adapter), for example, for a car it is a change from 76.98\% to 96.91\%, and for a cat from 48.82\% to 97.71\% (see Section 4).

---

### Official Review · Reviewer_ZPuh · 2025-10-31

**Soundness:** 2
**Presentation:** 3
**Contribution:** 2
**Rating:** 4
**Confidence:** 3

**Summary:**

The paper proposes a LoRA-guided model for controlling the unlearning process, dubbed UnGuide. The approach (i) trains LoRA adapters to forget a target concept $c$ by assigning it a matching concept $c_m$, and (ii) applies an adaptive control rule at the inference stage that mixes predictions from the baseline and LoRA-adapted models depending on the stability of noise suppression in early steps. Empirically, UnGuide is evaluated on object deletion (CIFAR-10) and explicit content removal (NSFW) (I2P) and exhibits lower NSFW detection rates than most baselines while maintaining reasonable FID/CLIP scores on MS-COCO dataaset.

**Strengths:**

- The authors reframe the removal of the specific concept as LoRA-based adaptation with a linear target in the noise prediction space, rather than as architectural changes or prompt rewrites. This leads to a simple but effective unlearning framework.
- The new metric is introduced to evaluate the performance of unlearning. It is referred to as the harmonic mean of effectiveness, specificity, and generality, and is meaningful and clear.
- UnGuide enables controllable concept erausre across different datasets, while CLIP/FID remains competitive compared to conventional algorithms.
- The paper is well-written and easy to understand

**Weaknesses:**

- The main concern is the insufficiently justified design of the loss function for unlearning. The objective pushes the noise prediction of the LoRA-guided model for the forbidden concept c toward a linear combination of the predictions of the original model, as given by eq. (4). In the article, the authors do not explain why this geometry in the noise space should be optimal for unlearning, nor why linearity (as opposed to other divergences or constraints) is appropriate.
- The prompt-conditional guidance is based on a norm gap statistic of early diffusion steps and seeds, compared to a neutral prompt reference, and then switches to negative/positive guidance accordingly. The paper includes some visualizations in the appendix, but there is little quantification of false positives/negatives or stability across time steps.
- The visualizations are informative, but there is a lack of analysis when the method fails: e.g., in the case of excessive suppression of harmless attributes or semantic drift for non-targets. Furthermore, as shown in Figures 16 to 26, the proposed UnGuide often produced oversaturated images, which is generally considered a limitation of guidance-based image generation.

**Questions:**

- How did you choose $\gamma$ and how sensitive are the results to it?
- What is the compute overhead of the multi-seed probing per prompt?
- Please clarify whether training separate adapters corresponds to training a single adapter with multi-target prompts. A brief experiment could be decisive for practical application.

---

> ### Author Response · Authors · 2025-11-27
>
> **W1**: Our loss design follows a standard principle used in diffusion models: linear manipulation of noise predictions. This geometry is well-established and directly parallels classifier-free guidance (CFG), see Eq. (1). This operation moves the model along the semantic direction.  We use the same mechanism but with a negative sign, applying the direction $(\varepsilon_p - \varepsilon_m)$ to push the model away from the forbidden concept by calculating: $\varepsilon_m - \gamma (\varepsilon_p - \varepsilon_m)$. The loss in Eq. (4) simply trains the LoRA adapter so that $\varepsilon_n \approx (\varepsilon_m - \gamma(\varepsilon_p - v_m))$. Thus, our formulation remains a standard linear manipulation of noise predictions, which is a widely used and theoretically grounded technique in diffusion models.
>
> **W2**: Tab. 8 shows that the norm-gap is stable both between different seeds and between denoising steps. For $\text{repeats} \geq 10$, the norm values ​​practically stabilize, and the variability between seeds remains small. Furthermore, the norms increase consistently with the number of denoising steps $t$, confirming the predictable and monotonic behavior of this metric. For $\text{repeats} \geq 10$, the dependencies between target, neutral, and non-target prompts are also consistent, which prevents false positives and false negatives. This allows for stable guidance switching. We also added additional ablation in Tab. 3  where we can see the high efficiency of our choosing guidance, achieving an average value close to 100\%. For each prompt, 50 independent estimates of the mean norm were made. The table illustrates how often the resulting mean maintained a correct relationship to the neutral norm.
>
> **W3**: We thank the reviewer for highlighting the importance of analyzing UnGuide's failure modes, including excessive suppression of harmless attributes and semantic drift in non-target concepts, as well as oversaturation in generated images. While UnGuide dynamically modulates the influence between the base and LoRA-adapted models via the UnGuidance mechanism to minimize unintended content removal, the model can occasionally produce outputs with semantic drift or over-suppression due to inherent challenges in disentangling overlapping features within the latent space. This behavior aligns with the instability observed during concept erasure, where diverse, unconstrained generative outputs may arise, as described in Section 3.
>
> Regarding oversaturation, as visible in Figures 18-28, this stems from the strong guidance, which can amplify pixel intensities and color saturation, a known trade-off in guidance-based synthesis that prioritizes targeted concept suppression. In the revised version, we acknowledge these limitations and plan to investigate refined guidance scaling techniques and image post-processing strategies in future work to improve image fidelity while maintaining unlearning effectiveness.

---

> > ### Author Response · Authors · 2025-11-27
> >
> > **Q1**: Thank you for your pertinent question about how to choose the **negative guidance** value. We tested several settings (1, 2, 3), and for objects, we chose $\gamma= 2$, as this value provides the best compromise between effective unlearning and overall generation stability. We performed additional ablation focusing on two classes (cat and automobile) and calculated metrics for $\text{Acc}_e$, $\text{Acc}_s$, $\text{Acc}_g$, and $\text{H}_o$. The results for the two tables below show that the method is stable across the entire range - the metrics change only slightly, and all three $\gamma$ values ​​lead to similar $\text{H}_o$ results. At $\gamma= 1$, unlearning weakens, and at $\gamma= 3$, unlearning strengthens, but this may affect specificity (automobile class).  $\gamma= 2$, which we use in this work, provides the best compromise between forgetting efficiency  (low $\text{Acc}_e$ and $\text{Acc}_g$) and maintaining generation quality for the remaining classes (high $\text{Acc}_s$). Additional ablation results are presented in Tab. 4.
> >
> > Importantly, the model is still stable in the range [1,3], which in effect shows that our method is not very sensitive to the choice of this parameter, and the value  $\gamma= 24$ proved to be the most balanced.
> >
> > Table for cat:
> >
> > | $\gamma$ | $\text{Acc}_e$ ↓ | $\text{Acc}_s$ ↑ | $\text{Acc}_g$ ↓ | $\text{H}_o$ ↑ |
> > |------------|---------------------|--------------------|--------------------|-----------|
> > | 1          | 2.43                | 98.62              | 4.34               | 97.27     |
> > | 2          | 2.98                | 98.80              | 2.66               | 97.71     |
> > | 3          | 2.25                | 98.55              | 3.55               | 97.58     |
> >
> > Table for automobile:
> > | $\gamma$ | $\text{Acc}_e$ ↓ | $\text{Acc}_s$ ↑ | $\text{Acc}_g$ ↓ | $\text{H}_o$ ↑ |
> > |------------|---------------------|--------------------|--------------------|-----------|
> > | 1          | 1.45                | 98.05              | 5.82               | 96.89     |
> > | 2          | 1.83                | 97.95              | 5.32               | 96.91     |
> > | 3          | 1.40                | 88.04              | 2.30               | 94.53     |
> >
> >
> > **Q2**: Tab. 8 shows that the norm-gap statistic stabilizes already at $\text{repeats} \geq 10$, regardless of the choice of time step $t$. In practice, a much cheaper variant can be used, e.g., $t=10$ with standard 50-step sampling, which significantly reduces the cost without sacrificing detection quality.
> >
> > Additionally, as shown in Fig. 15 and Fig. 16, partially denoised latents (e.g., after 10 steps) can be used directly as starting points for image generation, thereby avoiding the need for an additional full denoising pass.
> >
> > **Q3**: In principle, multiple concepts can be removed using a single LoRA adapter trained with multi-target prompts. However, this approach is less practical: LoRA modules are lightweight, and training separate adapters preserves modularity, allowing independent detachment or recombination (as demonstrated in Fig. 10 and Fig. 11). For these reasons we treat each concept as an independent LoRA module, which we find to be more flexible in real applications.

---

### Meta-Review · Area_Chair_32Yy · 2026-01-03

**Summary:**

The paper was reviewed by 4 experts and the initial reviews were 2422. The reviewers'  concerns are listed in the below field. Overall, there were still major concerns, such as motivation of the design (besides heuristic design rules), incremental novelty, quality of the result, and additional experiments. Overall, the paper is not quite ready for publication and requires a rewrite.

**Reviewer Concerns:**

**Reviewer ZPuh**
1. needs justification of the loss function for unlearning, as compared to other alternative losses.
2. little quantification of false positives/negatives or stability across time.
3. lack of analysis when the method fails. produces oversaturated images.

The AC thinks that Points 1 and 3 were not addressed convincingly. For Point 1, the response did not provide enough details in the justification -- why should it be based on CFG, rather than other types of losses? Regarding Point 3, it seems the concept removal severely affects the quality of the other generated classes, e.g., the quality of the deer changes significantly in Fig 18. However, there is no experiment to judge the visual quality.

**Reviewer K6aQ**
1. poor organization and paper flow.  Inconsistent notation.
2. overstated novelty: distilling negative guidance is already explored in ESD (Erasing Concepts from Diffusion Models, Rohit Gandikota, Joanna Materzynska, Jaden Fiotto-Kaufman, David Bau), while using LORA (rather than direct fine-tuning) is largely an implementation detail.
3. the adaptive guidance is computationally expensive and motivation is unclear.
4. missing analysis on key scaling parameters.

The AC thinks that Point 2 is critical, and the novelty is incremental.  Meanwhile, the reviewer raised a lot of issues with motivation of design choices (Point 3), which the authors answer that these are required to make it work (e.g., the "the unlearned model is often too weak, so additional repulsion is needed"). The method appears to be not that robust.

**Reviewer uC8E**
1. missing ablation study to base LoRA approach without unguidance (dynamic switching).
2. missing ablation study on adversarially chosen prompts.
3. how does unguided avoid distorting unrelated concepts?

The AC thinks that Point 3 was not addressed well - authors do not provide an analysis about why the the mean norm-based measure would minimize the risk of altering unrelated concepts.

**Reviewer Y2JL**
1. Limited novelty, extending AutoLoRA.
2. dependance on mapping content, possibly reducing diversity.
3. Should try on larger / recent models (SD_XL, SD3)
4. Run full benchmark as in MACE (erasing 200 celebrities)
5. stronger justification for design choice of AutoGuidance is needed.

The AC thinks that Point 2 and 4 were not addressed well enough. The additional experiments presented for Point 4 seem to be preliminary results.

**Reviewer Scores:**

The AC thinks that none of the reviewers would have changed their scores above marginal reject, since each has a significant point that needs further clarification.

---

### Decision · Program_Chairs · 2026-01-26

Reject